# Leveraging Weak Cross-Modal Guidance for Coherence Modelling via Iterative Learning

### Yi Bin
Tongji University
Shanghai, China
National University of Singapore
Singapore
yi.bin@hotmail.com

### Junrong Liao
University of Electronic Science and
Technology of China
Chengdu, China
charliel114514@gmail.com

### Yujuan Ding*
The Hong Kong Polytechnic University
Hong Kong SAR, China
dingyujuan385@gmail.com

### Haoxuan Li
University of Electronic Science and
Technology of China
Chengdu, China
lhx980610@gmail.com

### Yang Yang
University of Electronic Science and
Technology of China
Chengdu, China
yang.yang@uestc.edu.cn

### See-Kiong Ng
National University of Singapore
Singapore
seekiong@nus.edu.sg

### Heng Tao Shen
Tongji University
Shanghai, China
University of Electronic Science and
Technology of China
Chengdu, China
shenhengtao@hotmail.com

## ABSTRACT

Cross-modal coherence modeling is essential for intelligent systems to help them organize and structure information, thereby understanding and creating content of the physical world coherently like human-beings. Previous work on cross-modal coherence modeling attempted to leverage the order information from another modality to assist the coherence recovering of the target modality. Despite of the effectiveness, labeled associated coherency information is not always available and might be costly to acquire, making the cross-modal guidance hard to leverage. To tackle this challenge, this paper explores a new way to take advantage of cross-modal guidance without gold labels on coherency, and proposes the Weak Cross-Modal Guided Ordering (WeGO) model. More specifically, it leverages high-confidence predicted pairwise order in one modality as reference information to guide the coherence modeling in another. An iterative learning paradigm is further designed to jointly optimize the coherence modeling in two modalities with selected guidance from each other. The iterative cross-modal boosting also functions in inference to further enhance coherence prediction in each modality. Experimental results on two public datasets have demonstrated that the proposed method outperforms existing methods for cross-modal coherence modeling tasks. Major technical

modules have been evaluated effective through ablation studies. Codes are available at: *https://github.com/scvready123/IterWeGO*.

## CCS CONCEPTS

• **Information systems** → **Multimedia and multimodal retrieval**; **Retrieval models and ranking**.

## KEYWORDS

Cross-Modal Coherence Modelling, Iterative Learning

**ACM Reference Format:**
Yi Bin, Junrong Liao, Yujuan Ding, Haoxuan Li, Yang Yang, See-Kiong Ng, and Heng Tao Shen. 2024. Leveraging Weak Cross-Modal Guidance for Coherence Modelling via Iterative Learning. In *Proceedings of the 32nd ACM International Conference on Multimedia (MM '24), October 28-November 1, 2024, Melbourne, VIC, AustraliaProceedings of the 32nd ACM International Conference on Multimedia (MM'24), October 28-November 1, 2024, Melbourne, Australia.* ACM, New York, NY, USA, 10 pages. https://doi.org/10.1145/3664647.3681677

## 1 INTRODUCTION

Humans understand the world, to comprehend its semantics and dynamics, based on their organization and structuring of the information they perceive from various sources, seeking patterns and sequences that provide coherence to their understanding. In general, humans have remarkable skills in coherence modeling regardless of the information modality, which might be acquired from some childhood activities such as Picture-Telling [16, 44]. Such abilities are also essential to Artificial Intelligence (AI) models, which not only enhance the decision making by providing insights into the relationships and dependencies between different pieces of information [37], but also enable systems to understand context better [62],

*The corresponding author.

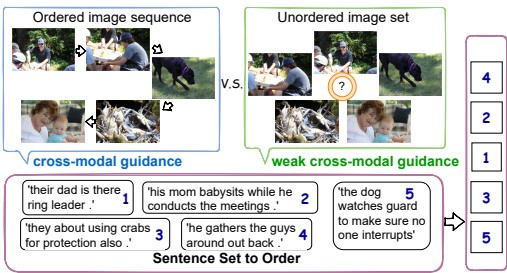

**Figure 1: Uni-modal and Cross-Modal (CM) Coherence Modeling (CM) tasks shown by a sentence ordering case according to whether using the guidance from another modality. This paper focuses on the CMCM task without using cross-modal ORDER information but only leverages weak guidance across modalities as shown in the green part.**

leading to more contextually relevant responses and actions. Coherence modeling may further benefit down-stream applications such as storytelling, educational materials and content marketing.

For these significance, coherence modeling has been an active research topic in AI research. For example, in the area of natural language processing (NLP), sentence ordering [4, 14, 38, 49] aims to organize a set of sentences into a coherence piece of text with a logically consistent order, which is one of the fundamental textural coherency modeling problems. Coherence modeling is also a keen topic in the era of Large Language Models (LLMs) since it has been involved in various areas, from the text summarization in NLP [62] to the video understanding in Computer Vision (CV) or even other multi-modal tasks [18, 40, 57, 61]. Existing effort has been made to investigate the coherence modeling capability of existing LLMs in different tasks, and demonstrated that state-of-the-art LLMs like GPT-4 have not achieved human-level ability in terms of coherence evaluation [52], leaving coherence modeling under-explored.

Coherence modeling is challenging due to the complex and diverse relations existing to determine the logic for the construction of different pieces of content [11, 21], including temporal relations, casual relations, discourse relations and many others. For example, in the sentence ordering task, it is required to first understand the semantics of each sentences, based on that to discover relations existing between different sentences and predict the order accordingly. From another perspective, exploring coherency patterns based on only single-modality data is difficult due to the limited information from only a single view, making the underlying logic difficult to explore [34, 41]. In comparison, Human perception is internally cross-modal, which facilitate them to better capture the dependency between different segments of content in any modality [28, 41]. Inspired by cross-modal human cognition [10, 36, 46], a previous study [9] proposed cross-modal coherence modeling method, which leverages additional information from another modality and use it as the reference to learn the order in target modality. The cross-modal correspondence and the order information in the additional modality may offer useful guidance. However, despite of the effectiveness, this method requires gold labels for the order information from the additional modality, which is not always available and would be expensive and time-consuming to acquire.

To further address this limitation, this paper introduces a more flexible cross-modal coherence modeling approach, which leverages the predicted instead of labeled order information in one modality to enhance the order modeling in another, as shown in the bottom part of Figure 1. The basic idea is that the two modalities for cross-modal coherence modeling are semantically correlated, if there is coherence information observed in one modality, the other modality could benefit from it as long as the coherence information is correlated and reliable. To be specific, we introduce the **We**ak Cross-modal **G**uided **O**rdering model (**WeGO**), which employs the predicted pairwise order in one modality as the reference information to guide the order prediction in another modality. Such weak guidance is selectively applied based on the prediction confidence of the reference order and the semantic correlation between the specific elements across modalities. An **Iter**ative Learning paradigm is further designed to optimize the ordering models of two modalities jointly, one at each time with feasible guidance from another, making the proposed method to be **IterWeGO**. At the inference stage, cross-modal order guidance functions iteratively for different modalities for multiple steps to boost uni-modal order prediction. The main contributions of this paper are as follows:

- We work on cross-modal coherence modeling task and propose a novel cross-modal guidance method WeGO, which effectively take advantage of relevant but weak order reference from another modality to guide the order modeling in the target modality.
- We propose an iterative learning paradigm to leverage weak cross-modal order guidance for both modalities during training, and learn the ordering models of two modalities jointly.
- We evaluate the proposed method on two public cross-modal coherence modeling datasets and compare it with existing competitive baselines. Experimental results demonstrate that the proposed method consistently outperforms all compared methods by large margins, showing great superiority of it.

## 2 RELATED WORKS
### 2.1 Coherence Modeling
Coherence modeling has been studied in various tasks in literature, including the most representative two: **Sentence Ordering** in NLP and **Visual Storytelling** in multimedia understanding.

**Sentence ordering** is representative coherence modeling task in the area of NLP, with early-stage methods mostly built based on domain knowledge and language-based features [4, 8, 19, 27, 39]. For example, these methods use vectors of linguistic features in probabilistic transition models. In recent years, with the development of deep learning and NLP technologies, more advanced methods apply an encoder-decoder framework and retrieve the final order using pointer networks [14, 26, 43, 53, 58]. Topological sorting is also applied to address the sentence ordering problem [47, 49]. With the boost of pre-trained language models, BERT-based [15], BART-based [13] and other pre-trained language model-boosted sentence ordering networks have been proposed and achieved competitive performance. Graphs are applied to enhance the sentence representations for the sake of ordering learning [59, 63], also to implement set modeling without orders and more suitable for the input.

Storytelling is a higher-level coherence modeling task for coherency modeling, which requires to generate the sentences that

can be ordered properly to construct a coherent story [30, 33]. **Visual storytelling** is a further extension of storytelling with multimodal input, which requires the generated story to be consistent to the given visual content [1, 29]. Such a cross-modal coherence modeling task help to develop human-like artificial intelligent methods that understand the grounded event structure that go beyond descriptive language only [42]. Early methods mostly leveraged CNNs and RNNs into an integrated encoder-decoder architecture to comprehend visual input and generate textual output respectively [12, 55]. More recent methods attempted to enhance the reasoning or planing abilities of the model by applying more advanced or large-scale pre-trained modules. For example, Liu et al. [42] proposed to introduce the planning procedure for visual storytelling relying on pre-trained language models, which help to generate less repetitive, more logical stories with more details. Although both visual storytelling and our cross-modal ordering task target at cross-modal coherence modeling, we are faced with unordered materials for both modalities while the visual storytelling offers well-ordered image sequences as grounding reference for the text generation.

## 2.2 Cross-Modal Learning

Cross-modal learning generally aims to learn a unified representation of two modalities, mostly vision and language. It plays an important role in many different cross-modal tasks, such as vision-language retrieval [6, 31, 56], vision question answering [3, 48, 60], vision-language alignment [9], multi-modal sentiment classification [24, 32], visual captioning [5, 7], cross-modal relation extraction [17, 20], etc. With the extraordinary development of the large-scale pre-training, Vision-Language pre-training (VLP) has become a mainstream research direction recently. The goal of VLP is to learn unified representation of different modalities through large-scale learning based on large-scale image-text pair data, which can further benefit the above-mentioned downstream tasks. With a dual-stream architecture, CLIP has exhibited remarkable performance on zero-shot recognition and several downstream tasks by applying effective contrastive learning [51]. For further enhancement, PyramidCLIP [23] introduced fine-grained interactions between two modalities to achieve better cross-modal alignment. CyCLIP [25] introduced geometrical consistency constraints. SoftCLIP [22] relaxed the one-to-one cross-modal alignment constraint and instead introduced intra-modal guidance to enable many-to-many relationships between the two modalities, which further enhance the pre-training performance cross-modal learning model.

Cross-modal learning is generally a sub-field of multi-modal machine learning [2, 45, 54]. From another perspective, it aims to use data from additional modalities to improve a uni-modal task. Lin et al. [41] proposed a cross-modal learning method by treating data from other modalities as additional training samples to help with the uni-modal few-shot learning. Jin et al. [31] proposed to apply the idea of Banzhaf Interaction to explicitly capture the fine-grained semantic relationships between vision and text, and use it as additional learning signals to improve the contrastive cross-modal learning. Qin et al. [50] investigated how to improve the correspondence across visual and textural modalities in cross-modal learning, which might contain ubiquitous noise in labeled pair data.

Ju et al. [32] proposed joint learning method for sentiment analysis by incorporating the image-text relation to leveraging the visual guidance for text understanding. With a similar idea, cross-modal coherence modeling task aims to leverage the additional guidance from another modality to facilitate the coherence model learning in one modality. However, different with previous work, this study tries to leverage not only the cross-modal semantic information, but also the order information to specifically help with the uni-modal coherence modeling task.

## 3 METHOD: IterWeGO

In this section, we introduce the proposed **We**ak Cross-modal **G**uided **O**rdering model with **Iter**ative Learning (**IterWeGO**). In the following part, We first introduce single modal encoders for the semantic and context encoding of multi-modal input. Then, we detail the cross-modal guidance module to leverage weak order as reference. In the end, we demonstrate the iterative boosting mechanism for learning and inference, which enables to enhance ordering learning and prediction in two modalities simultaneously.

## 3.1 Problem Formulation

This paper studies the cross-modal coherence modeling, specifically on the task of sentence and image ordering problem. Given a set of sentences and a set of images, both unordered, the task is to learn to organize the elements in each set with proper orders based on the semantics so that the two parts of content would make one coherent story, either in natural language or visual illustration.

## 3.2 Intra-Modal Semantic & Contextual Encoding

Given an image set $\mathcal{X}_i$ consisting of several unordered images $\{X_{i1}, X_{i2}, ...X_{iM}\}$, where $N = |\mathcal{X}_i|$ denotes the number of images in the set. Each image $X_{ik} \in \mathcal{X}_i$ can be analyzed by a visual model $F$, i.e., a semantic encoder to extract the visual content features as $\boldsymbol{e}_{ik} = F_\sigma(X_{ik})$, where $\sigma$ denotes the parameters in the model $F()$. It is important to exploit semantic relations between different elements, i.e., images or sentences, for recovering coherence in the set. Towards this end, we further design an intra-modal context encoder equipped with the scaled dot-product self-attention to obtain the contextual representation [8]. Context-aware representations for the set of images can therefore be obtained as $\hat{\boldsymbol{E}}_i = MH\_Att_\beta(\boldsymbol{E}_i)$, where $\boldsymbol{E}_i$ is the matrix composed of the content representations of all images in the set as $E_i = \{e_{i1}, e_{i2}, ...e_{i|\mathcal{X}_i|}\}$. $\beta$ denotes the parameters in the context encoder $MH_Att()$. Similarly, for the input sentence set $\mathcal{S}_j = \{S_{j1}, S_{j2}, ..., S_{jN}\}$, where $M = |\mathcal{S}_j|$, each sentence $S_{jk}$ can be analyzed by a semantic encoder $G_\eta$ first to obtain the content representation $\boldsymbol{e}_{jk}$. The whole set of content representations is further passed to a context encoder $MH\_Att_\gamma$ similar as images to obtain context-aware representation $\hat{\boldsymbol{E}}_j$. $\eta$ and $\gamma$ are trainable parameters for the encodes.

## 3.3 Semantic-Aligned Cross-Modal Order Guidance

Based on the context-aware representations, we can further model the relative order of any two elements (in the textual or visual

—

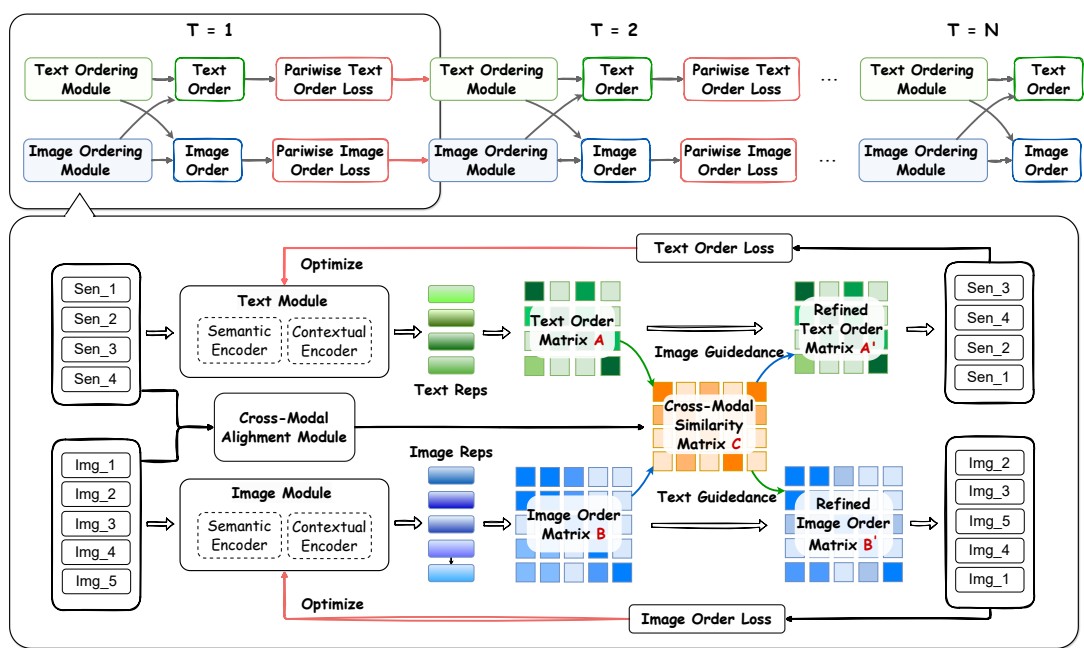

**Figure 2: Framework of the proposed IterWeGO model. An iterative learning paradigm is designed to optimize the ordering models of two modalities jointly with continuous guidance from each other. The weak cross-modal order guidance is applied selectively at each learning step based on the predicted pairwise order through semantic cross-modal alignment.**

modality) by devising a pairwise order classifier $h_{\delta^I}(x_1, x_2)$ with trainable parameters $\delta^I$. For example, for two images $X_{i1}, X_{i2}$ with context-aware representations being $\hat{\boldsymbol{e}}_{i1}, \hat{\boldsymbol{e}}_{i2}$, their pairwise order result turns to be $o_{i_{12}} = h_{\delta^I}(\rho(\hat{\boldsymbol{e}}_{i1}, \hat{\boldsymbol{e}}_{i2}))$, where $\rho$ is the integration function, for example, can be the concatenation operation. We formulate pairwise order prediction as a binary classification problem and apply a logistic function further to predict the probability of two situations ($X_{i1} > X_{i2}$ or $X_{i2} > X_{i1}$) as $s_{i_{12}} = \text{Sigmoid}(o_{i_{12}})$.

For a target image set $\mathcal{X}_i$, there are in total $\binom{n}{2}$ pairs. All pairwise order probability scores of the set consist of an $N \times N$ square, corresponding to the number of images in the set, which can be denoted as $B \in \mathbb{R}^{N \times N}$. Similarly, for each sentence set we have the order matrix $A \in \mathbb{R}^{M \times M}$ in which each element denotes the pairwise order probability of the corresponding two sentences.

To explore deeper connections between two modalities and enhance the semantic alignment between the associated image and text sets, we further leverage **cross-modal semantic similarity modeling** as illustrated in Figure 2. Specifically, we apply a pretrained cross-modal models on the input sets of the two modalities respectively, and further obtain the cross-modal similarity matrix $C \in \mathbb{R}^{M \times N}$. Each element $c_{mn} \in C$ measures the semantic similarity of the two corresponding elements from two modalities based on their cross-modal content representations.

With the cross-modal semantic similarity matrix, we can apply cross-modal order guidance based on a simple idea that the prediction of pairwise order in one modality is highly related to the corresponding pairwise score in another modality. The cross-modal similarity matrix serves as a bridge to link two modalities to ensure the order guidance is only applied between **semantic-aligned**

cross-modal elements. Furthermore, our cross-modal guidance is only invoked when the prediction confidence of the pairwise score in the referred modality is high, making it qualified to become a guidance. To effectively leverage cross-modal order guidance, we design a cross-modal guided order matrix updating (CGO-MU) algorithm to update the pairwise order matrix in one modality. For example, the order matrix of the image set $B$ can be refined with the CGO-MU algorithm based on the order matrix of the sentence set $A$ and the cross-modal similarity matrix $C$ as $B' = \text{CGO-MU}(B|A, C)$. Detailed steps of CGO-MU is introduced in Algorithm 1, according to which we can also derive the way to obtain the refined sentence order matrix by $A' = \text{CGO-MU}(A|B, C)$.

### 3.4 Cross-Modal Learning and Inference with Iterative Boosting

As illustrated in the upper part in Figure. 2, we introduce an **I**terative **B**oosting (IB) mechanism to enable the cross-modal guidance to be applied and benefit the ordering model learning in both the visual and textual modalities. The iterative learning paradigm is applied to further optimize the uni-modal modules of image and text ordering iteratively with cross-modal guidance being leveraged and updated along with the iterative training. The optimization is implemented based on the cross-entropy loss on the pairwise order prediction, which is specifically as follows if taking image ordering learning as an example:

$$\mathcal{L}_{Img} = -\frac{1}{|\mathcal{X}|} \frac{1}{|\mathcal{X}_i|^2} \sum_{\mathcal{X}_i \in \mathcal{X}} \sum_{k,l \in \mathcal{X}_i} y_{kl}\text{Softmax}(b'_{kl}), \qquad (1)$$

**Algorithm 1:** Cross-modal Guided Order Matrix Updating (CGO-MU) for Image Matrix

---

**Input** : Sentence Set Order Score Matrix $A$, Image Set Order Score Matrix $B$, Cross-Modal Semantic Similarity Matrix $C$, Mask Threshold $\theta$

**Output** : Refined Image Set Score Matrix $B'$

---

1 Masked Sentence Set Score Matrix $\hat{A}$
2 $\hat{A} = \text{zeros\_like}(A)$
3 **Step 1**:
4 **for** $i \in [0, M]$ **do**
5    **for** $j \in [0, M]$ **do**
6       **if** $A[i, j] > \theta$ **then**
7          $\hat{A}[i, j] = A[i, j]$
8       **end**
9    **end**
10 **end**
11 $B' = \text{copy}(B)$
12 **Step 2**:
13 **for** $p \in [0, M]$ **do**
14    **for** $q \in [0, M]$ **do**
15       **if** $\hat{A}(p, q) \neq 0$ **then**
16          $idx_1 = \text{Argmax}(C(p))$
17          $idx_2 = \text{Argmax}(C(q))$
18          $B'(idx1, idx2) += A(p, q)$
19       **end**
20    **end**
21 **end**
22 **RETURN** $B'$

---

where $b'_{kl} \in B'$ is refined predicted pairwise order of the elements $X_k$ and $X_l$ (not that for simplicity we omit $X$ and denote images by their index as $k$ and $l$ in the above equation). $y_{ij}$ is the ground-truth label of the corresponding order, which could be 0 or 1. $\mathcal{X}$ is the collection of all image sets in the training data. All training parameters involved in the images ordering model ($\sigma$, $\beta$ and $\delta^I$) get updated after the optimization. Similarly, the sentence ordering loss can be defined as:

$$\mathcal{L}_{Sen} = -\frac{1}{|\mathcal{S}|} \frac{1}{|\mathcal{S}_i|^2} \sum_{\mathcal{S}_i \in \mathcal{S}} \sum_{k,l \in \mathcal{S}_i} z_{kl} \text{Softmax}(a'_{kl}), \tag{2}$$

where $a'_{kl} \in A'$ and $z_{kl}$ are the refined predicted and ground-truth pairwise order of the element pair $< S_k, S_l >$ respectively. Optimizing $\mathcal{L}_{Sen}$ updates all parameters involved in the sentence ordering process.

In inference, we use topological sort algorithm to find the order of the whole image/sentence set based on the pairwise order predicted by our cross-modal coherence model. Specifically, we build a ordering graph (directed) for each set, in which each element (image or sentence) serves as a node and the pairwise order suggests the directed edge between the two corresponding nodes. For each node, we can calculate its overall order score by summarizing the in/out edges, and sort the score of all nodes to obtain the final order list for the target set. Iterative boosting functions during inference too, which enables to update the order prediction iteratively across modalities for several times to achieve better accuracy for ordering in each modality.

# 4 EXPERIMENTS AND RESULTS

To evaluate the effectiveness of the proposed approach, we conduct extensive experiments on two standard coherence modeling datasets, *i.e.*, SIND [30] and TACoS-Ordering [9].

## 4.1 Experimental Settings

We apply ViT and BERT as the content encoder to process images and sentences respectively, and conduct fine-tuning on two uni-modal ordering tasks. The contextual encoders for text and image share the same structure, which is combined by Self Attention Layers and Feed Forward Layers [9]. The cross-modal alignment module relying on semantic similarity is implemented by a CLIP model. This model is fine-tuned before applying in the IterWeGO with the cross-modal pairs in the target dataset in a contrastive learning manner. Threshold to select valid cross-modal guidance is set to 0.8 for image-to-text and 0.9 for text-to-image respectively based on empirical results. The setting of threshold is a trade-off between the quality and quantity of the leveraged cross-modal guidance. Empirical results suggest that the performance of the model stays stable when it is set within a certain range. During training, we employ Adam [35] optimizer to minimize the loss, with initialized learning rate of $2 \times 10^{-4}$. The batch size of SIND is set to 64, while TACoS-Ordering is 32. We apply 8 parallel heads for all the multi-head attention layer and the hidden size is 768. In inference, we apply multi-step cross-modal guidance to enhance the order prediction in each modality untill the best performance is reached. In accordance with previous work [8, 9], we apply three metrics to evaluate the performance of different methods for cross-modal ordering tassk, which are Accuracy (short in Acc), Perfect Match Ratio (PMR) and Kendall's Tau ($\tau$).

## 4.2 Comparison with Baselines

We compare the proposed IterWeGO model with the following competitive coherence modeling approaches.

- **LSTM+PtrNet [43]:** This method uses an LSTM-based encoder to obtain the context representation and the pointer network as the decoder to predict the order.
- **L-TSort [49]:** It implements topological sort to optimize the relative order constraint between paired sentences.
- **AttOrderNet (AON) [14]:** It applies self-attention-based encoder and autoregressive decoder, which is very similar to our work except the autoregressive decoding. Following [9], we test two versions, AON-UM and AON-CM, for comprehensive comparison. AON-UM exploits single-modal data only, which is just as applied in the original paper. AON-CM is the expanded cross-modal version which takes data from both modalities as input.
- **RankNet [38]:** It also applies similar encoders with AON and our NACON, but employs ranking strategy for the output order. We also implement two versions of single-modal and cross-modal, RankNet-UM and RankNet-CM, for comparison.
- **NACON [9]:** It is the state-of-the-art coherence modeling method employing a basic encoder-decoder framework with an unique non-autoregressive decoder to tackle with the permutation challenge for order prediction. It leverages cross-modal guidance with ground-truth orders to boost the coherence modeling of the singe modality.

**Table 1: Overall Cross-Modal Coherence Modeling Performance**

| Model | Sentence Ordering | | | | | | Image Ordering | | | | | |
|---|---|---|---|---|---|---|---|---|---|---|---|---|
| | SIND | | | TACoS-Ordering | | | SIND | | | TACoS-Ordering | | |
| | Acc | PMR | $\tau$ | Acc | PMR | $\tau$ | Acc | PMR | $\tau$ | Acc | PMR | $\tau$ |
| LSTM+PtrNet | 43.84 | 11.26 | 0.4471 | 38.86 | 10.09 | 0.6967 | 23.23 | 2.27 | 0.0714 | 23.34 | 2.41 | 0.4543 |
| L-TSort | 42.66 | 10.03 | 0.4798 | 38.1 | 9.44 | 0.7009 | 23.13 | 1.97 | 0.072 | 20.93 | 1.93 | 0.4089 |
| AON | 44.16 | 11.97 | 0.4494 | 39.42 | 10.61 | 0.7003 | 24.54 | 2.65 | 0.0906 | 23.5 | 2.76 | 0.4355 |
| RankNet | 42.33 | 9.79 | 0.4587 | 34.91 | 8.46 | 0.6701 | 22.75 | 1.66 | 0.078 | 19.15 | 1.54 | 0.3618 |
| NACON-UM | 47.86 | 14.48 | 0.5115 | 43.06 | 11.22 | 0.7405 | 25.12 | 2.59 | 0.1056 | 24.95 | 3.24 | 0.4641 |
| NACON-UM* | 50.81 | 16.87 | 0.5498 | 44.42 | 11.93 | 0.76 | 27.84 | 3.68 | 0.1584 | 26.92 | 3.36 | 0.5169 |
| IterWeGO-UM | 49.48 | 15.39 | 0.5646 | 44.47 | 11.99 | 0.7665 | 26.37 | 2.41 | 0.1601 | 29.52 | 3.84 | 0.5987 |
| IterWeGO-UM-NoCon | 48.19 | 15.21 | 0.5552 | 45.07 | 12.24 | 0.77 | 33.55 | 6.68 | 0.3094 | 29.85 | 3.72 | 0.6342 |
| IterWeGO (Ours) | **54.45** | **20.91** | **0.6450** | **46.20** | **12.52** | **0.7775** | **36.48** | **8.01** | **0.3512** | **32.92** | **4.61** | **0.6579** |
| Improv. (%) | 7.2 | 23.9 | 17.3 | 4.0 | 4.9 | 2.3 | 31.0 | 117.7 | 121.7 | 22.3 | 37.2 | 27.3 |
| NACON[1] | 65.06 | 35.49 | 0.6549 | 45.39 | 11.37 | 0.7668 | 44.05 | 13.9 | 0.3371 | 28.14 | 3.54 | 0.5197 |
| NACON* | 72.92 | 48.09 | 0.7318 | 54.49 | 30.73 | 0.8309 | 48.03 | 17.23 | 0.4433 | 40.84 | 15.17 | 0.7282 |

[1] NACON and NACON* both leverage additional cross-modal order labels, therefore can be seen as the upper bound of our model. NACON* is an advanced version applying same visual and textual encoders as IterWeGO.

## 4.3 Overall Performance

The overall performance of all compared methods, including the proposed IterWeGO and the baselines, are shown in Table. 1. From the table we have the following observations.

- Compared to the methods without using additional cross-modal order labels, the proposed IterWeGO method outperforms them all on both the Sentence Ordering and the Image Ordering tasks across the two datasets, showing consistent effectiveness in the general coherence modeling task. We can also observe that the leveraged cross-modal guidance (in Sec. 3.3) and the contextual encoding (in Sec. 3.2) are both effective by comparing the performance of IterWeGO with two variants: -UM and -UM-NoCon.
- The proposed IterWeGO gains more performance improvement in the image ordering task than for the sentence ordering task. From the overall results we can see that all methods perform better for sentence ordering than image ordering, which suggests that sentences are relatively easier to be ordered than images, and images ordering are more challenging. Since our IterWeGO leverages weak cross-modal guidance, which is predicted relative order information instead of the pre-obtained gold labels, it is more effective when the modality to receive the guidance is challenging and the modality to offer the guidance is easy. Image ordering with sentence order guidance is exactly such a task, in which case the weak guidance is more reliable. In comparison, offering image order guidance for sentence ordering tries to apply a weaker model to facilitate a stronger model, which therefore is more difficult. Nevertheless, our IterWeGO still achieves significant performance improvement comparing other methods, showing general effectiveness of the idea of leveraging cross-modal order guidance to boost the uni-modal ordering.
- By comparing different metrics, we further observe that our method gain relatively more performance in terms of PMR and $\tau$, which measure exact match and relative orders respectively. Such results demonstrate good capability of our IterWeGO to

**Table 2: Performance of IterWeGO variants on SIND with different *Iterative Boosting (IB)* settings: 1) without IB in Training and Inference; 2) without IB in the Inference; 3) without IB in Training; and 4) the IterWeGO.**

| Model | Sentence Ordering | | | Image Ordering | | |
|---|---|---|---|---|---|---|
| | Acc | PMR | $\tau$ | Acc | PMR | $\tau$ |
| w/o IB in T&I | 49.48 | 15.39 | 0.5646 | 26.37 | 2.41 | 0.1610 |
| w/o IB in T | 50.92 | 17.09 | 0.5891 | 34.79 | 6.50 | 0.3214 |
| w/o IB in I | 49.11 | 14.80 | 0.5659 | 24.68 | 2.45 | 0.1232 |
| IterWeGO | 54.46 | 20.91 | 0.6450 | 36.48 | 8.01 | 0.3512 |

explore the orders rather than the exact positions of different segments in both modalities.

- Table 1 also demonstrates the noticeable difference of performance of IterWeGO across datasets. The effectiveness of our model seems to be more obvious on the SIND dataset than TACoS-Ordering, suggesting that the idea of cross-modal order guidance is particularly useful when the images and sentences are better aligned and matched.
- We also illustrate the performance of two NACON models at the bottom two rows in the table. Specifically, NACON denotes original method [9] and NACON* denotes the advanced version using ViT and BERT to replace the image and sentence encoders in the original model to keep consistent with the IterWeGO method. Both NACON and our IterWeGO incorporate cross-modal order guidance. Although the performance of NACONs seems to be higher, it needs to be noticed that NACONs use gold order labels while our method only relies on predicted relative order information, which is much weaker guidance comparatively. From this perspective, NACONs may be seen as upper-bound reference to our IterWeGO model. While compared to NACONs, our model is also more feasible to apply since it does not need such gold labeled order annotations.

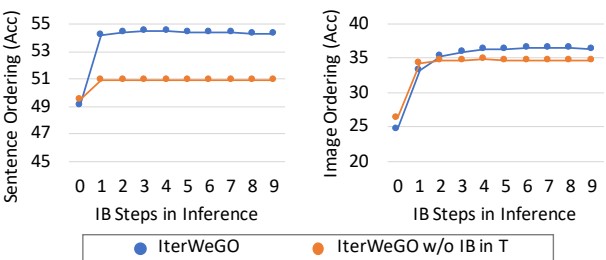

Figure 3: Illustration of the performance of two type of models (IterWeGO and IterWeGO w/o IB in training) with different Iterative Boosting (IB) steps during inference.

## 4.4 Impact of Iterative Boosting

To verify the effectiveness of our iterative boosting strategy, we conduct ablative experiments on SIND dataset and illustrate the results in Table 2. The first two rows of results correspond to the models without applying cross-modal iterative boosting during training. As shown from the results, applying iterative inference only can boost the ordering performance. When we implement iterative boosting strategy during both training and inference stages, the performance achieves the best owing to the synchronization signal between training and inference. We also note an interesting phenomenon that solely apply iterative training yields a slightly lower result comparing to without using it (*w/o IB in I* v.s. *w/o IB in T&I*). This may result from the discrepancy between training and inference, because in the case of *w/o IB in I*, the inference is conducted in uni-modal only, which is different from the training stage leveraging cross-modal guidance. But if the cross-modal guidance, aka, the iterative boosting is also applied during inference, the performance will significant improve (*w/o IB in I* v.s. *IterWeGO*).

We take a further step to investigate the mechanism of our iterative strategy by delving into the iteration steps during inference. Specifically, we implement 10 iteration steps during inference and measure the performance for each step. The variation trend across steps is shown in Figure 3. From the figure, we observe that all the methods gain significant improvements at the first iteration, and gradually reach the plateau values. The cross-modal training setting (our IterWeGO) exhibits boosting for more steps and finally outperforms the uni-modal version by a large margin, which verifies the effectiveness of the iterative strategy during both training and inference stages.

## 4.5 Impact of Cross-Modal Guidance

One important technical part in the proposed IterWeGO is to leverage the weak relative order guidance across modalities to enhance the uni-modal ordering learning. Although the overall performance of IterWeGO has been demonstrated superb, it is still not clear whether the cross-modal order guidance contributes to the overall performance and how does it contribute. To answer these questions, we test variants of IterWeGO models and compare their performance together with IterWeGO. The first variant is IterWeGO-FFE

Table 3: Performance of IterWeGO variants with different cross-modal enhancement on SIND. FFE: Feature Fusion Enhancement; ROE: Relative Order Enhancement.

| Model | Sentence Ordering | | | Image Ordering | | |
|---|---|---|---|---|---|---|
| | Acc | PMR | $\tau$ | Acc | PMR | $\tau$ |
| IterWeGO-FFE | 44.49 | 11.85 | 0.4886 | 29.13 | 3.84 | 0.2211 |
| IterWeGO-ROE | 51.94 | 17.90 | 0.6154 | 34.13 | 6.09 | 0.3184 |
| IterWeGO | **54.45** | **20.91** | **0.6450** | **36.48** | **8.01** | **0.3512** |

Table 4: Performance of IterWeGO and LLMs (GPT) on two ordering tasks based on 176 testing samples from SIND.

| Model | Sentence Ordering | | | Image Ordering | | |
|---|---|---|---|---|---|---|
| | Acc | PMR | $\tau$ | Acc | PMR | $\tau$ |
| GPT-UM | 53.41 | 26.70 | 0.5784 | 25.34 | 3.41 | 0.0910 |
| GPT-CM | 43.75 | 17.05 | 0.4795 | 30.23 | 5.11 | 0.2250 |
| IterWeGO | 55.23 | 31.25 | 0.6625 | 37.39 | 8.52 | 0.3591 |

(Feature Fusion-Enhanced), which applies cross-modal enhancement by leveraging **semantic guidance** through cross-modal feature fusion. The second variant is IterWeGO-ROE (Relative Order-Enhanced), which leverages **order guidance** instead of semantic guidance. Compared with IterWeGO, IterWeGO-ROE use the pre-trained CLIP model to measure the cross-modal correlations, which is the basis to apply cross-modal guidance, while IterWeGO applies a fine-tuned CLIP model, whose semantic correlation measurement has been adapted properly to the target datasets we use.

Corresponding experimental results have been illustrated in Table 3, from which we have the following observations. First, our IterWeGO outperforms two variants on both tasks across to datasets, showing that our current design of cross-modal guidance enhancement is generally better than the other two. We also notice that IterWeGO-ROE consistently outperforms IterWeGO-FFE, which demonstrates that applying cross-modal guidance with relative order information is significantly more effective than semantic enhancement via simple feature fusion. Furthermore, with delicate fine-tuning for the cross-modal alignment module, our IterWeGO achieves marked performance improvement compared to IterWeGO-ROE. Since the cross-modal alignment module serves as the bridge to pass the specific guidance across modalities, it partly determine the quality of the leveraged guidance as well as the effectiveness of the guidance enhancement, which thereby significantly impact the overall performance of the whole model.

## 4.6 Case Study

To straightforwardly illustrate the effectiveness and mechanism of our IterWeGO model, we exhibit three cases in Figure 4, two for the comparison between different methods and one to show the iterative updating for ordering during inference. From case (a), we can observe that when the uni-modal ordering suffering from the lack of coherence information, it cannot recover the order both for texts and images. NACON [9] employs cross-modal positional attention to leverage the gold order information from the other modality to

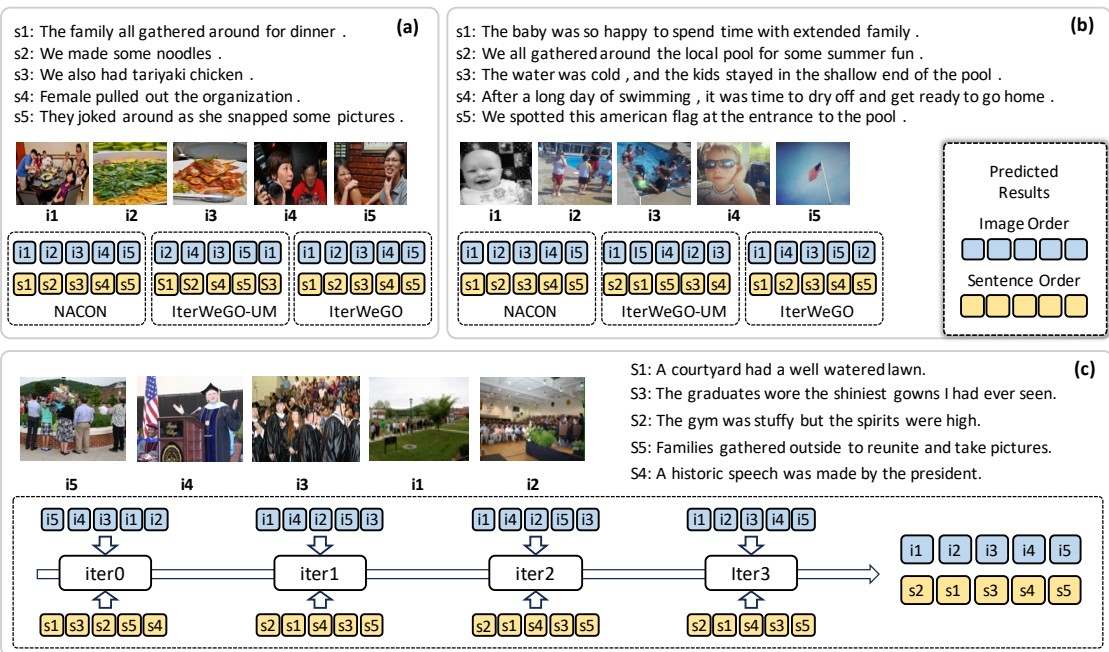

**Figure 4: Three case illustration. (a) and (b) show the image and sentence orders predicted by our IterWeGO and the key baseline NACON and the variant IterWeGO-UM. (c) illustrates the iterative updating process for two ordering tasks with cross-modal guidance during inference by our IterWeGO.**

assist the target modality and achieves much better results, which could play as an upper bound. While our IterWeGO only leverage weak guidance from the other modality, it also recovers the gold order with iterative boosting paradigm. Even IterWeGO cannot totally recover the gold order, it still benefits from the guidance of relative order hidden in the other modality, and partially recover the order. With such partial recovering, the PMR will not improve but Accuray and $\tau$ will gain improvements.

As discussed in Section 4.4, the iterative boosting strategy significantly improve the order recovering. To make the mechanism of iterative boosting more straightforwardly, we illustrate an example of iterative inference in case (c). From this case, we observe at step 0 (unimodal inference), the order is far from coherent, but based on the weak cross-modal guidance at step 1, several sentences and images can be placed at the right position. With the iteration going, more elements, $i.e.$, images and sentences, are placed at the ground-truth positions, and results in recovering the gold order in the end. In summary, these cases directly illustrate the effectiveness and superiority of our proposed IterWeGO model.

### 4.7 Comparison with Large Foundation Models

As large foundation models, $e.g.$, large language models (LLMs) and large multimodal models (LMMs), have been demonstrating superb capability in various areas and tasks. Here we take an attempt to investigate the LMMs for ordering problem with weak cross-modal guidance. We random sample 200 cases from the test set and prompt GPT-4 (for pure text ordering) and GPT-4V (for image and cross-modal ordering) to recover the order of a set of elements (images or sentences). We note that GPT rejected 24 samples for ordering,

resulting in 176 valid sample for final comparison. As the results shown in Table 4, to our surprise, GPT achieves inferior ordering performance comparing with our IterWeGO. More interesting, with the weak guidance from the other modality, GPT-CM yields worse performance than the unimodal scenario (GPT-UM) for text ordering, but gains improvement for image ordering. Bin $et~al.$ [8] pointed out that the reason of GPT fails in ordering problem may come from the autoregressive nature of GPT. We go a step further to hypothesize that the reason for the inferior performance in the cross-modal scenario may be because GPT processes all content as input, summarizing the common topics of images and sentences, but fails to capture intra-modal coherence. Anyway, the prompting engineering is also a reason of unexpected performance of GPT, we will keep improving the prompts to yield better performance.

## 5 CONCLUSIONS

This paper presented a novel cross-modal coherence modeling method that focuses on taking advantage of cross-modal unordered guidance to enhance the order learning in single modality, named as Weak Cross-modal Guided Ordering (WeGO). More specifically, it selectively employs the predicted relative order information in one modality to guide corresponding order modeling in another based on proper semantic alignment. Furthermore, to facilitate the ordering model learning in both modalities, an iterative learning diagram was proposed. Extensive experiments done in two public datasets demonstrated the effectiveness of the proposed method. Two major technical components, namely weak cross-modal guidance and iterative boosting in both training and learning, have been evaluated to be effective with solid experiments and anslysis.

# ACKNOWLEDGMENTS

This work is supported by the National Natural Science Foundation of China under grant 62102070 and 62220106008, and Sichuan Science and Technology Program under grant 2023NSFSC1392. This research is supported by A*STAR, CISCO Systems (USA) Pte. Ltd and National University of Singapore under its Cisco-NUS Accelerated Digital Economy Corporate Laboratory (Award I21001E0002).

We also sincerely thank all the ACs and reviewers for their efforts on our work and appreciate the useful comments for improving it.

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
