# OpenReview forum: "Leveraging Weak Cross-Modal Guidance for Coherence Modelling via Iterative Learning"
_acmmm.org/ACMMM/2024/Conference — MM2024 Poster_

### Official Review · Reviewer_md3V · 2024-05-24

**Rating:** 4
**Confidence:** 3

**Summary:**

The paper presents a method for improving cross-modal coherence modeling, particularly in tasks such as sentence and image ordering. The core idea is to utilize weak cross-modal guidance, where predicted pairwise order information from one modality (e.g., text or images) helps guide the coherence modeling in another modality without relying on gold-standard labeled data, which can be costly and time-consuming to obtain.

Overall, the paper provides a robust framework for enhancing cross-modal coherence modeling by integrating weak guidance from correlated modalities and refining predictions through iterative learning.

**Strengths:**

1. The paper is well-structured and clearly explains the methodology, experimental setup, and results. Figures and tables effectively illustrate key points, enhancing readability.
2. The WeGO model introduces a novel method of leveraging weak cross-modal guidance without requiring costly labeled data, addressing a significant challenge in coherence modeling.
3. The paper evaluates the WeGO model on two public datasets (SIND and TACoS-Ordering), demonstrating improvements over baseline methods

**Limitations:**

1. The model depends on the quality of the predicted pairwise orders from the other modality. If these predictions are inaccurate, the overall performance can degrade significantly. The paper could benefit from a discussion on mitigating this dependency.
2. The iterative learning and inference processes, while effective, introduce significant computational complexity.
3. The method relies on cross-modal semantic similarity, which may not always be robust, especially in cases where the modalities do not align well semantically.
4. Lack of comparison with similar methods. The paper could be strengthened by comparing WeGO with other weakly-supervised or semi-supervised coherence modeling approaches.

**Suitability:**

3

---

### Official Review · Reviewer_uCMZ · 2024-05-24

**Rating:** 4
**Confidence:** 3

**Summary:**

This paper proposes a Weak Cross-Modal-guided Coherence Modelling framework via Iterative Learning, leveraging an unsupervised idea without labeled orders to align the elements in two different modalities.

**Strengths:**

1. This paper is easy to read and understand and its logic is clear.
2. The idea seems innovative.
3. Main experimental results validate the effectiveness of the proposed method.

**Limitations:**

1. The motivation illustrated in Figure 1 needs deeper analysis.
2. The expression of ‘modelling’ and ‘modeling’ should be consistent.
3. Although the proposed method does not need prior labels to support the optimization process, it is not clearly described in the method or experiments whether the pre-trained models in line 339 were obtained based on prior labels or not. If the pre-trained model is label-based, I think the proposed label-unneeded strategy is less innovative.
4. Line 100 needs concrete literature support.
5. More recently published SOTA baselines are needed.
6. Some grammar issues. For example, ‘understand and create’ in line 10 should be ‘understanding and creating’.

**Suitability:**

3

---

### Official Review · Reviewer_z5Mo · 2024-05-24

**Rating:** 5
**Confidence:** 3

**Summary:**

The article presents a novel method for cross-modal coherence modeling called the Weak Cross-Modal Guided Ordering (WeGO) model. This approach leverages weak guidance from high-confidence predicted orders in one modality to enhance coherence modeling in another, through an iterative learning paradigm. This method operates without requiring labeled coherence information, making it advantageous for scenarios where such data is scarce or expensive to obtain.

**Strengths:**

1.Innovation: Introduces a unique method of using predicted, rather than labeled, order information for cross-modal coherence, increasing accessibility and reducing costs.
2.Iterative Learning Paradigm: Enhances both the learning and inference stages, allowing continuous improvement of coherence modeling.
3.Empirical Validation: The method outperforms established baselines and shows effectiveness through rigorous experimental validation.

**Limitations:**

1. The model's effectiveness may be limited by the accuracy of initial order predictions in one modality.
2. The iterative nature of the model could complicate implementation and increase computational overhead

**Suitability:**

3

---

### Official Review · Reviewer_6Bfe · 2024-05-25

**Rating:** 3
**Confidence:** 2

**Summary:**

Cross-modal coherence modeling is crucial for intelligent systems to help users organize and structure information, enabling them to understand and create content from the physical world coherently, much like humans. Previous work on cross-modal coherence modeling attempted to use sequential information from another modality to assist in recovering coherence in the target modality. While effective, the availability of labeled relevant coherence information is not always guaranteed, and the acquisition cost may be high, making cross-modal guidance difficult to utilize. To address this challenge, this paper explores a new approach utilizing cross-modal guidance and proposes the Weak Cross-Modal Guidance Ordering (WeGO) model. This model leverages high-confidence predictions of paired sequences from one modality as reference information to guide coherent modeling in another modality. Furthermore, an iterative learning paradigm is designed to jointly optimize the consistency modeling of the two modalities through mutually selected guidance. Iterative cross-modal enhancement also plays a role in the reasoning process to further enhance coherence predictions within each modality.

**Strengths:**

The paper introduces a novel cross-modal guidance method, WeGO, effectively utilizing weak but relevant sequential references from another modality to guide sequential modeling in the target modality.

The paper presents an iterative learning paradigm, leveraging weak cross-modal sequential guidance from two modalities during the training process, jointly learning the ordering models of the two modalities.

The paper thoroughly evaluates the proposed method on two public cross-modal coherence modeling datasets.

**Limitations:**

1. On line 382, ‘the prediction of pairwise order in one modality is highly related to the corresponding pairwise score in another modality.’ Is there any reference or theoretical basis for this conclusion? If so, could the authors please provide a more detailed explanation?

2. On line 386, ‘confidence of the pairwise score in the referred modality is high, making it qualified to become a guidance.’ How is the confidence of the pairwise score designed and determined? Could the authors provide further discussion on this?

3. In the experimental section, it is mentioned that ‘Threshold to select valid cross-modal guidance is set to 0.8 for image-to-text and 0.9.’ What would happen if other values were chosen for the threshold? Could the authors further discuss this through experimentation?

4. Has the author considered adding more datasets to validate the method?

5. With the exception of NACON, the baseline methods mentioned in the experimental section are primarily based on work from 2018-2020. Are there any updated approaches in this field, for instance, methods proposed in 2023, that could be considered as baselines? If so, should the authors consider incorporating these approaches for comparison to further demonstrate the effectiveness of the proposed method?

**Suitability:**

3

---

### Meta-Review · Area_Chair_4fCh · 2024-07-11

**Recommendation:** Accept (Poster)
**Confidence:** 4

**Metareview:**

**Conclusion: Accept as a Poster Paper**

After thorough consideration of the review reports from the independent reviewers, I conclude that this paper is recommended accepting as a Poster Paper for the ACM MM 2024 conference. The decision is based on the innovative approach proposed by the authors, which contributes significantly to the field of cross-modal coherence modeling, despite certain limitations that require further investigation.

**Strengths:**

1. **Innovative Approach**: The WeGO model introduces a novel method of using weak cross-modal guidance from one modality to enhance coherence modeling in another. This is particularly beneficial in scenarios where labeled coherence information is scarce or expensive to obtain.
2. **Iterative Learning Paradigm**: The model leverages an iterative learning process, which not only optimizes consistency modeling across modalities but also continuously refines predictions, enhancing overall performance.
3. **Empirical Validation**: The proposed method has been rigorously validated on two public datasets (SIND and TACoS-Ordering), demonstrating its effectiveness and improvements over baseline methods.

**Weaknesses:**

1. **Dependency on Initial Predictions**: The model's effectiveness heavily relies on the quality of the initial predicted pairwise orders from one modality. Inaccurate predictions can significantly degrade performance.
2. **Computational Complexity**: The iterative learning and inference processes introduce additional computational overhead, which could complicate implementation and practical application.
3. **Comparison with Recent Methods**: The paper lacks a comprehensive comparison with more recent state-of-the-art (SOTA) methods, particularly those proposed in the last few years. Including such comparisons would strengthen the paper's claims of effectiveness.

In summary, while there are some concerns regarding computational complexity, dependency on initial predictions, and the need for broader comparisons, the innovative approach and empirical validation of the WeGO model make this paper a valuable contribution to the field. Therefore, I recommend its acceptance as a Poster Paper to encourage further discussion and exploration of its potential applications and improvements.